# Effects of Strength Training on Cross-Country Skiing Performance: A Systematic Review

**DOI:** 10.3390/ijerph19116522

**Published:** 2022-05-27

**Authors:** Arkaitz Castañeda-Babarro, Paula Etayo-Urtasun, Patxi León-Guereño

**Affiliations:** Health, Physical Activity and Sports Science Laboratory, Department of Physical Activity and Sports, Faculty of Education and Sport, University of Deusto, 48007 Bizkaia, Spain; paula.etayo@opendeusto.es (P.E.-U.); patxi.leon@deusto.es (P.L.-G.)

**Keywords:** cross-country skiing, cross-country skiers, strength training, resistance training, performance

## Abstract

Traditionally, cross-country skiing has been known for having a strong endurance component; however, strength demands have significantly increased in recent years. Given this importance, several studies have assessed the effects of strength training in cross-country skiing. Therefore, the aim of this systematic review was to analyze the results of those studies. A detailed search of four databases (Pubmed, Scopus, Web of Science and Cochrane Library) was conducted until February 2022, according to the Preferred Reporting Items for Systematic Reviews and Meta-Analyses statement. Ten eligible studies were selected from the 212 records identified, all of them including young well-trained skiers and interventions of 6–12 weeks. Results showed that maximal strength training may improve some important variables: for instance, performance, double-poling economy and maximal strength. However, this type of training failed to change other indicators such as peak oxygen consumption. Concurrent training, which combines endurance and maximal strength training, seems to be effective to improve performance. The mechanisms responsible for the improved economy of double poling might be due to a lower percentage of maximal strength during double poling at a given workload, which could increase performance. Future studies should include longer interventions which analyze a more varied sample.

## 1. Introduction

Cross-country skiing has been an Olympic sport since the first Winter Games in Chamonix (France) in 1924 [1,2]. Over the last few decades, more effective training and advances in equipment and track preparation have led to an increase in the speeds of cross-country skiers [2,3]. In addition, new techniques, race formats and distances have been incorporated. Nowadays, international competitions involve both freestyle and classic techniques, and different formats are observed, including pursuit, mass-start and relays. Furthermore, the distances of the different events have a wide range, from 1.2 km sprint races to 50 km long-distance races [3,4].

Traditionally, cross-country skiing has had a strong aerobic component and, consequently, the training regime of cross-country skiers has been based on endurance training [3]. Therefore, the aerobic capacity of cross-country skiers is very important, including maximal oxygen uptake (VO_2_max) and peak oxygen uptake (VO_2_peak) [5]. However, due to the changes previously mentioned, anaerobic demands have increased in several cross-country skiing events, as the last sprint is decisive [3]. Sunde et al. [6] demonstrated that maximal upper body strength is a determinant factor in double poling performance. In addition, it has been proved that work economy is a decisive factor in cross-country skiing, as it offers a great advantage in saving energy and thus increasing speed [7,8]. Therefore, it has been suggested that inter-individual differences in exercise economy might be related to differences in training with regard to maximal strength training and speed training [9].

In this context, different types of strength training have been studied in relation to endurance sports. According to Rønnestad and Mujika [10], maximal strength training can be effective for improving endurance performance. This is in line with the meta-analysis by Ambrosini et al. [11], who found that strength training was useful for increasing performance in different endurance sports, including cross-country skiing. Furthermore, Berryman et al. [12] concluded that the implementation of additional strength training could be effective to enhance middle- and long-distance performance by improving the energy cost of locomotion, maximal power and maximal strength.

Regarding studies that assessed the effects of strength training in cross-country skiers, the results obtained to date do not show the same direction. On the one hand, some studies found that incorporating maximal strength training into traditional endurance training can lead to important improvements related to cross-country skiing performance [13,14]. As an example, Hoff et al. [15] found significant improvements in maximal strength, double poling economy and performance after nine weeks of maximal upper body strength training in well-trained females. Moreover, Losnegard et al. [16] reported significant improvements in aerobic parameters such as VO_2_max. On the other hand, various studies did not observe significant improvements in VO_2_max after a period of strength training [17,18]. Furthermore, Carlsson et al. [19] compared strength training versus ski ergometer training and found no significant between-group differences for gross efficiency, upper-body strength or time to exhaustion (TTE). Therefore, the effect of strength training on cross-country skiing remains unclear.

Taking into account the different results found in the literature, the aim of this systematic review is to gather and analyze studies that assessed the effects of strength training on different variables related to performance in cross-country skiing.

## 2. Materials and Methods

This article is a systematic review of existing studies involving investigation of the effects of strength training in cross-country skiing athletes. This systematic review was carried out following the guidelines summarized in the Preferred Reporting Items for Systematic reviews and Meta-Analyses (PRISMA) statement [20]. 

### 2.1. Search Strategy

The search for articles was performed with an end date of 23 February 2022. Articles were collected from the following databases: PubMed, Scopus, Web of Science and Cochrane Library. The phrase with which we found the largest number of valid articles to include in the review was as follows: (“strength training” OR “resistance training”) AND (“cross country skiing” OR “cross country skiers”). 

### 2.2. Eligibility Criteria

The inclusion criteria were defined using the PICOS model (Population, Intervention, Comparison, Outcome and Study type) [21]. Therefore, studies were included if they met the following criteria: (a) studies including only cross-country skiers (Population); (b) studies including a strength training intervention (Intervention); (c) studies including a control group performing traditional cross-country ski training (mostly aerobic training and some muscular endurance) (Comparison); (d) studies assessing variables related to cross-country performance; and (e) experimental studies (Type of study). After examining the inclusion and exclusion criteria, a total of 10 references were selected for this systematic review.

### 2.3. Data Collection and Selection Process

The search strategy consisted of collecting all results from the four databases and removing duplicates. Articles identified by the search strategy were screened by title and abstract. Then, a complete reading of the selected articles was carried out and articles that did not meet the inclusion criteria were excluded. These studies were ruled out for four different reasons: (I) other sports were included, (II) all subjects performed the same general strength training, (III) strength training was not performed, (IV) study protocol was insufficient.

### 2.4. Methodological Quality Assessment

After conducting the literature search and selection, an assessment of the methodological quality of the studies was performed using the Physiotherapy Evidence Database (PEDro) scale. According to Maher et al. [22], PEDro scale is a reliable 11-item scale designed to measure the methodological quality of randomized control trials (RCT). The studies were evaluated by qualitative assessment and not meeting the exclusion criteria. The methodological quality of the studies was ranked as high when the score ranged from 6 to 10 (the first item was not used to calculate the total score). The methodological quality of studies scoring between 4 and 5 points was considered moderate, and scores 3 or lower indicated low quality. The average of 5.5 overall quality indicated that the set of studies included in this systematic review had moderate quality. Moreover, none of the studies were considered to be of low quality (Table 1).

## 3. Results

### 3.1. Study Selection

A flow diagram of the literature search is presented in Figure 1. A total of 212 studies were found through searching the databases, of which 56 duplicate studies were removed. An additional 125 studies were excluded after analyzing their titles and abstracts. Among the 31 selected articles, 21 studies were excluded because they did not meet the inclusion criteria. Therefore, 10 studies remained for qualitative synthesis.

### 3.2. Study Characteristics

Table 2 shows the characteristics of the selected studies such as the characteristics of the subjects, the training regime, the testing procedures and the results obtained for the variables assessed. The studies in this systematic review included a total of 204 cross-country skiers. All the subjects were well-trained skiers, most of them being regional or elite competitors. In terms of age, most of the participants were young, with a mean age of 20.22 years. In addition, five studies included male skiers [13,14,17,18,24], two studies female skiers [15,23] and three studies mixed samples [16,19,25].

All studies conducted maximal strength training [13,14,15,16,17,19,23,24], except Mikkola et al. [18], which implemented explosive type strength training. However, this was not the only study that included explosive strength exercises in the training routine, as two other studies combined maximal and explosive strength exercises in the designed workouts [17,25]. The control groups performed traditional training regimes, which usually included muscular endurance training. In addition, five studies used upper body strength training [13,14,15,23,24], whereas the other five targeted the whole body [16,17,18,19,25].

All studies evaluated the effects of strength training on important variables in cross-country skiing performance. However, the variables examined were different between the studies. Of the studies, 70% assessed the impact of strength training on skiers’ maximal strength using one repetition maximum (1 RM) strength tests [13,14,15,16,17,19,24]. In addition, several studies directly analyzed the effects of resistance training on cross-country skiing performance, although they used different tests and parameters including TTE [13,14,15,19,24], maximal double poling speed [18], average power relative to body weight during the 5 min double poling test [16] and maximal power measured by the 20-s test [23] or by an incremental test on a double poling ergometer [25]. Some studies also examined the effects of strength training on double poling economy [13,14,15,19,23,24]. Finally, various studies examined some aerobic parameters, although they used different indicators. The most studied aerobic parameters were VO_2_max and VO_2_peak [13,14,15,16,17,18,19,23,24].

Regarding maximal strength, most of the studies that carried out 1 RM tests reported significant increases in the group that performed strength training, whereas no significant change was found in the control group [13,14,15,16,23]. Ofsteng et al. [24] was the only one in which no between-group differences were found.

In addition, five studies observed significant differences in performance on a double poling ergometer between groups after a strength training intervention [13,14,15,16]. Three studies indicated a significant improvement between pre- and post-test results, but they did not find significant differences when groups were compared [18,19,24,25]. The study by Skattebo et al. [23] was the only one in which no significant changes in performance were found, measured as power output on the 20-s test.

The effects of strength training on double poling economy were also contradictory. On the one hand, three studies reported significant improvements in economy between the strength training group and the muscular endurance training group [13,14,15]. On the other hand, two studies did not find significant within group improvements [19,23]. Finally, seven of the nine studies that examined changes in aerobic parameters found no significant differences in any indicator [13,14,15,17,18,19,24]. The only studies that found differences in aerobic parameters did so in VO_2_max. Skattebo et al. [23] found a significant reduction in VO_2_max in the strength training group, whereas Losnegard et al. [16] showed an improvement in VO_2_max after 12 weeks of strength training.

## 4. Discussion

The aim of this systematic review was to gather and analyze studies that assessed the effects of strength training on different variables related to performance in cross-country skiing. The main findings indicated that maximal strength training may be effective for improving some variables related to cross-country skiing, such as strength levels, work economy and performance. However, numerous studies did not find significant differences in VO_2_max and VO_2_peak, so further research is warranted.

According to some studies, maximal strength training can lead to significant improvements in double poling economy measured via a ski ergometer [13,14,15]. However, other studies failed to find significant differences when work economy was measured while roller skiing [16,19,23,24]. It has therefore been suggested that the apparatus used to measure economy may be the reason why the studies obtained different results, due to the similarity between the ergometer technique and strength exercises such as seated pull-down [16].

However, Mikkola et al. [18] showed significant differences in work economy, measured as steady-state oxygen consumption, while roller skiing. In contrast, Skattebo et al. [23] did not find reductions in O_2_ cost after upper body heavy strength training. It is important to underline that the mean power output of the subjects did not change significantly between rested and fatigued states. Therefore, the lack of reduction in O_2_ cost could be due to an insufficient load to induce performance decline, which might potentially mask the effects of strength training in a fatigued state [9].

Some studies indicated that improved work economy could lead to significant improvements in cross-country skiing performance [7,8]. This is in line with research conducted in other endurance sports, which found significant improvements in both economy and performance [10,12]. In general, most studies found that after several weeks of maximal strength training, the performance of cross-country skiers increased significantly [13,14,15,16,24,25]. Some of them reported large improvements in TTE, ranging from 47% to 136% [13,14,15]. However, some studies found no significant changes in submaximal O_2_ cost when additional strength training was compared with traditional training [23,25]. One of the reasons for this disagreement could be the different tests used to measure performance. Thus, some studies measured O_2_ uptake after only 2 min at submaximal workloads [13], whereas others used stages of 5 min in their tests [23]. Moreover, Skattebo et al. [23] found no significant differences in work economy due to increased neuromuscular fatigue, resulting in reduced performance. It may seem surprising that this study found no significant differences in performance given that it was the study that found the greatest improvements in maximal strength levels. It is important to note that although a certain level of strength is required, greater strength does not necessarily translate into improved performance, as endurance capabilities and technique are both crucial factors in minimizing fatigue accumulation [26,27]. Therefore, Skattebo et al. [23] may not have found an improvement in performance because VO_2_max was significantly reduced, or even because they did not focus enough on technique to enable athletes to translate strength gains into the complex movement of skiing.

Although several studies found significant improvements in double poling economy, most of the studies found no significant differences in aerobic parameters such as VO_2_max and VO_2_peak after a period of additional strength training [13,14,15,17,18,24]. This lack of change in VO_2_max and VO_2_peak after additional strength training has been previously observed in studies conducted in running, cycling and swimming [10,12]. The only study which found significant differences in VO_2_max during skate roller skiing was Losnegard et al. [16]. Their strength training intervention lasted 12 weeks, while the other studies had a maximum duration of 10 weeks. Therefore, interventions between 6 and 10 weeks could be relatively short for improving aerobic parameters in elite cross-country skiers [12], and future interventions should have a longer duration in order to assess the long-term effects of strength training.

Another reason that can partially explain this lack of change might be that further improvement in maximal strength is needed in order to improve perfusion conditions, which could positively affect VO_2_peak [13,15]. In addition, subjects were well-trained cross country skiers who had relatively high preintervention values, which makes it more difficult to improve VO_2_max levels [28]. Therefore, research should also be conducted in recreational skiers to examine if strength training leads to improved aerobic parameters in non-competitive skiers.

In addition, the volume and intensity of endurance training might not have been enough to improve those parameters, as most of the endurance training was performed below the anaerobic threshold [18]. Moreover, in most of the studies, endurance training was based on long distances with little high intensity work, which is what presumably induces the greatest improvement in VO_2_max [28].

These results may seem contradictory, as improvements in economy were observed without finding differences in VO_2_max and VO_2_peak. However, these results are consistent with previous research suggesting that inter-individual differences in economy can only be partially explained by differences in VO_2_peak [7,8].

It has been suggested that one factor that could contribute significantly to the improvement in double poling economy would be increased maximal strength [13,14,15,18,24]. Increased maximal strength might lead to a reduction in the percentage of maximal force involved in double poling, which might facilitate work economy through blood perfusion [6,13,15]. Moreover, it seems that it is the rate of force development and power production, rather than increased strength as such, that alters work economy [14,15,17]. Decreased TPF during double poling without changing the poling frequency at a standard workload results in longer rest periods between strokes and muscle contractions, which may enhance blood flow and thereby increase work economy [14,15,18]. Nevertheless, the effects of strength training on peripheral blood circulation have not been fully elucidated. Therefore, studies assessing the effects of maximal strength training on circulation should be carried out.

Most of the studies found significant differences in 1 RM before and after the intervention, although the magnitude of this change differed between investigations [13,14,15,16,23,24]. Thus, some studies observed an increase of 9.9–14.5% in 1 RM [13,15], whereas others found greater improvements [16]. The largest change was found by Skattebo et al. [23], who reported a 24% increase in 1 RM in skiers aged 16 to 18 years. Therefore, it is possible that the improvements were greater due to the subjects’ stage of maturation [29]. In addition, the participants attended a higher percentage of sessions (95–100%) than in other studies [13,15,24], which might have led to greater strength development.

The adaptations that led to an increase in maximal strength are unknown. Some studies found no significant differences in total body mass, and therefore authors suggest that the increase in maximal strength was due to neural adaptations [13,14,15,24]. However, other studies found significant increases in upper body lean mass and upper arm circumference and/or cross-sectional area (CSA) in triceps branchii muscle, suggesting that strength development could be partly explained by improved hypertrophy [16,18,23,24]. In this context, the study by Vahtra et al. [25] reported that increases in lean mass depended on the periodization of the training, suggesting that it is possible to prevent hypertrophy and thus improve work economy.

This systematic review has some limitations. Firstly, the studies used different tests to measure the variables included in this review, which makes it difficult to compare the results of different studies. Secondly, most studies showed a moderate overall quality in the qualitative assessment using the PEDro scale. Performing randomization seems to be difficult due to athletes’ and coaches’ preferences. In addition, maintaining double-blind protocols is not easy since skiers know the content of the training regime. Moreover, interventions lasted 6–12 weeks, so caution is advisable when long-term effects are considered. Lastly, only one study assessed the effects of additional explosive-type strength training, so more studies should investigate the impact of this type of strength training.

## 5. Conclusions

In conclusion, it seems that strength training, especially maximal strength training, may enhance performance in cross-country skiing by improving work economy. Apparently, a certain level of strength is important for overcoming fatigue, although gains in maximal strength do not always induce improvements in economy and performance. Thus, the rate of strength development becomes particularly relevant. The results suggest that maximal strength training could contribute to these improvements without causing a reduction in aerobic capacity, as long as volume and intensity are adjusted to the subject so that adaptations occur without excessive fatigue.

This systematic review may have several practical implications for coaches and athletes. Firstly, it may be advisable to perform maximal strength training, depending on the needs and characteristics of the skier. It is important to underline that the strength exercises must be movement-specific in order for strength gains to translate into improved performance. On the other hand, frequency, volume and intensity must be correctly programmed so that adaptations occur without excessive fatigue, as this might have a negative impact on performance.

## Figures and Tables

**Figure 1 ijerph-19-06522-f001:**
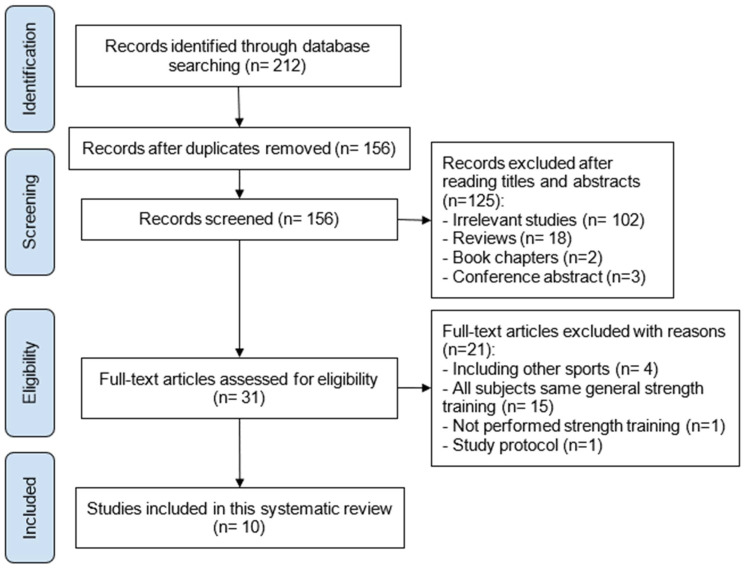
Flow diagram for the identification, screening, eligibility and inclusion of studies.

**Table 1 ijerph-19-06522-t001:** Methodological quality assessment using PEDro scale.

Study	1	2	3	4	5	6	7	8	9	10	11	Total	Overall Quality
Paavolainen et al. [17]	Y	N	N	Y	N	N	N	Y	Y	N	Y	4	Moderate
Hoff et al. [15]	N	Y	N	Y	N	N	Y	Y	Y	Y	Y	7	High
Hoff et al. [13]	Y	Y	N	Y	Y	Y	N	Y	Y	Y	Y	8	High
Østerås et al. [14]	Y	Y	N	Y	N	N	N	Y	Y	N	Y	5	Moderate
Mikkola et al. [18]	Y	N	N	Y	N	N	N	Y	Y	Y	Y	5	Moderate
Losnegard et al. [16]	Y	N	N	Y	N	N	N	Y	Y	Y	Y	5	Moderate
Skattebo et al. [23]	Y	N	N	Y	N	N	N	Y	Y	Y	Y	5	Moderate
Carlsson et al. [19]	Y	Y	N	Y	N	N	N	Y	Y	Y	Y	6	High
Øfsteng et al. [24]	Y	N	N	Y	N	N	N	Y	Y	Y	Y	5	Moderate
Vahtra et al. [25]	Y	N	N	Y	N	N	N	Y	Y	Y	Y	5	Moderate
Mean	-	-	-	-	-	-	-	-	-	-	-	5.5	Moderate
Median	-	-	-	-	-	-	-	-	-	-	-	5	-

Y: yes, N: no. 1: Eligibility criteria were specified. 2: Subjects were randomly allocated to groups. 3: Allocation was concealed. 4: The groups were similar at baseline regarding the most important prognostic indicators. 5: There was blinding of all subjects. 6: There was blinding of all therapists/researchers who administered the therapy/protocol. 7: There was blinding of all assessors who measured at least one key outcome. 8: Measures of at least one key outcome were obtained from more than 85% of the subjects that were initially allocated to groups. 9: All subjects for whom outcome measures were available received the treatment or control condition as allocated or, where this was not the case, data for at least one key outcome were analyzed using “intention to treat”. 10: The results of between-group statistical comparisons were reported for at least one key outcome. 11: The study provided both point measures and measures of variability for at least one key outcome.

**Table 2 ijerph-19-06522-t002:** Characteristics of studies included in review.

Study	Participants	Training Intervention	Testing Procedures	Outcomes	Results
Paavolainen et al. [17]	15 well-trained male (18–21 years) 7 IG/8 CG	6 weeks. IG: 34–42% explosive and heavy STR and 58–66% endurance training. CG: 15% endurance strength and 85% endurance.	- SJ and CMJ - Isometric strength tests - Ski-walking on a treadmill	Heights SJ and CMJ Maximal isometric strength TPF VO_2_max Thaer Than	↑ ↔ ↓ ↔ ↔ ↔
Hoff et al. [15]	15 well-trained female (17–18 years) 8 IG/7 CG	9 weeks. Additional training: IG: 2 h/w general STR + 7 h/w maximal STR CG: 13 h/w general STR General STR: >20 rep, <60% of 1 RM Maximal STR: 3 × 6 RM, 85% of 1 RM	- 1 RM strength test (pull-down exercise) - TTE test on ski ergometer - TTE test on running treadmill	Cdp Relative workload Than 1 RM TTE TPF VO_2_max VO_2_peak	↑ ↓ ↔ ↑ ↑ ↓ ↔ ↔
Hoff et al. [13]	19 well-trained male (15–23 years) 9 IG/10 IG	8 weeks. Additional training: IG: 3 sessions per week, pull-down exercise, 3 × 6 85% of 1 RM CG: <85% of 1 RM	- 1 RM strength test (pull-down exercise) - VO_2_max test on treadmill running - VO_2_peak test on ski ergometer - TTE test on ski ergometer	1 RM PF at 2 submaximal loads TPF TTE Cdp Body weight VO_2_max VO_2_peak	↑ ↑ ↓ ↑ ↑ ↔ ↔ ↔
Østerås et al. [14]	19 well-trained male (19–23 years) 10 IG/9 IG	9 weeks. Additional training: IG: 3 times per week, pull-down exercise, 3 × 6 85% RM CG: <85% of 1 RM.	- 1 RM strength test (pull-down exercise) - Graded protocol on ski ergometer - TTE test on ski ergometer	Cdp Relative strength 1 RM TTE Peak power production Movement velocity VO_2_peak Than	↑ ↑ ↑ ↑ ↑ ↑ ↔ ↔
Mikkola et al. [18]	19 well-trained male (19–27 years) 8 IG/11 CG	8 weeks. IG: 27% of total training volume was explosive-type STR. 6–10 exercises, 2–3 × 6–12	- Concentric and isometric strength tests - 30 m DP test with roller skis - MAST - Ski-walking on a treadmill - Cdp and maximal aerobic capacity tests	Maximal isometric force Vmax VO_2_max Maximum IEMG Aerobic capacity Steady-state oxygen	↔ ↑ ↔ ↔ ↔ ↓
Losnegard et al. [16]	19 well-trained male and female (18–27 years) 9 IG/10 CG	12 weeks. IG: 1–2 additional strength sessions. Half squat, seated pull-down, standing double-poling and triceps press. 3 × 3–10 RM	- CMJ - Work economy and VO_2_max during roller skiing - VO_2_max during running - 100 m sprint skiing test - 1 RM strength tests - DP performance - Rollerski time-trial	1 RM CMJ CSA in m. triceps brachii Upper body LBM DP performance VO_2_max roller skiing VO_2_max running Rollerski time-trial test performance	↑ ↔ ↑ ↑ ↑ ↑ ↔ ↔
Skattebo et al. [23]	16 well-trained female (16–18 years) 9 IG/7 CG	10 weeks. IG: 2 additional strength workouts. 3 exercises: seated pull-down, standing DP and triceps press. 3 × 4 RM − 10 RM	- 1 RM strength test (pull-down exercise) - DP sprint-test - DP finishing-test - VO_2_max running test	1 RM Performance Submaximal O_2_-cost VO_2_max VO_2_peak Upper arm circumference	↑ ↔ ↔ ↓ ↔ ↑
Carlsson et al. [19]	33 well-trained male and female (16–19 years) 14 IG/19 CG	6 weeks. IG: 2 additional sessions per week. 5 exercises (2 × 5 − 8 RM) + 3 core-stability exercises CG: 2 additional sessions of interval training on ski-ergometer	- Gross-efficiency test - Maximal-speed test - VO_2_peak test - Strength tests	1 RM MWR Gross efficiency TTE Vmax VO_2_peak	↑ ↑ ↔ ↑ ↑ ↔
Øfsteng et al. [24]	21 well-trained male (19–34 years) 21 IG/8 CG	8 weeks. IG: 3 additional strength sessions per week. 3 exercises: standing DP, seated pull down and triceps press. Weeks 1–3: 3 × (12–10–6) 2–3’r Weeks 4–6: 3 × (8–10–5) 2–3’r Weeks 7–8: 3 × (6–8–4) 2–3’r	- 1 RM strength test - Incremental TTE test while roller skiing - Submaximal DP followed by TTE test - Near-infrared spectroscopy	1 RM TTE after prolonged DP TTE diff VO_2_peak Body mass EMG parameters SmO2rest Upper-body LBM	↑ ↑ ↔ ↔ ↔ ↔ ↔ ↑
Vahtra et al. [25]	28 well-trained male and female (16–19 years) 8 IG1/10 IG2/10 CG	10 weeks. IG1 and IG2: additional strength training (IG1 2 times/w and IG2 1–3 times/w). 5–6 exercises. 2–3 × 4–10 RM CG: 2 times/w, >15 rep	- Pmax test on ski ergometer	Pmax in all groups Lean mass in IG2 Lean mass in IG1 and CG	↑ ↔ ↑

IG: intervention group, CG: control group, STR: strength training, h: hours, /w: per week, SJ: squat jump, CMJ: countermovement jump, TPF: time to peak force, VO_2_max: maximal oxygen consumption, Thaer: aerobic threshold, Than: anaerobic threshold, ↑: statistically significant increase, ↔: no statistically significant change, ↓: statistically significant reduction, RM: repetition maximum, TTE: time to exhaustion, Cdp: double-poling economy, VO_2_peak: peak oxygen uptake, PF: peak force, <: less than, >: more than, DP: double-poling, IEMG: integrated electromyography, MAST: maximal anaerobic skiing test, CSA: cross-sectional area, m.: muscle, LBM: lean body mass, O_2_: oxygen, MWR: mechanical work rate, Vmax: maximal speed, SmO2rest: baseline saturation levels, Pmax: maximal aerobic performance test.

## Data Availability

Not applicable.

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
