# Peer review of "Effects of Strength Training on Cross-Country Skiing Performance: A Systematic Review"

_ijerph, 2022, doi:10.3390/ijerph19116522_

Round 1

Reviewer 1 Report

The aim of this systematic review: „Effects of strength training on cross-country skiing perfor-2 mance. A systematic review“ is to gather and analyze the effect of strength training on different variables related to performance in cross country skiing.

The article is interestingly written and provides useful information for practice. The strength is the gathering of publications dealing with the topic, which is described and implemented with precision. However, there are facts that need to be improved. In particular, the second objective: the analysis of the effect of strength training is not, in my opinion, sufficiently described in the methodology, it should also be given more attention in the results section.

Major comments:

If the aim of the thesis is to analyze the effect of strength training and not to analyze publications dealing with the effect of strength training, it is necessary to supplement the thesis.

The work analyses in detail the quality of already published articles on the topic (Methodological Quality Assessment), but the methodology part does not specify what methodology was used for the analysis of the effect of strength training on different variables related to performance in cross country skiing.

Minor comments:

E.g. Line 38 and beyond: VO2max and VO2peak – I recommend using subscript:

Line 275: extra text: 5. Conclusions

Author Response

REVIEWER 1

The aim of this systematic review: Effects of strength training on cross-country skiing performance. A systematic review” is to gather and analyze the effect of strength training on different variables related to performance in cross country skiing.

The article is interestingly written and provides useful information for practice. The strength is the gathering of publications dealing with the topic, which is described and implemented with precision. However, there are facts that need to be improved. In particular, the second objective: the analysis of the effect of strength training is not, in my opinion, sufficiently described in the methodology, it should also be given more attention in the results section.

Major comments:

If the aim of the thesis is to analyze the effect of strength training and not to analyze publications dealing with the effect of strength training, it is necessary to supplement the thesis.

The work analyses in detail the quality of already published articles on the topic (Methodological Quality Assessment), but the methodology part does not specify what methodology was used for the analysis of the effect of strength training on different variables related to performance in cross country skiing.

AUTHOR´S

Thank you for your contributions. It is true that there is no calculation of the effect of resistance training on cross-country skiing performance. This is due to the large number of different variables used in the articles included in the review. More than one calculation would be necessary and, above all, given the number of variables, the conclusions would lose reliability, so it was decided to modify the objective and write it as follows:

“Taking into account the different results found in the literature, the aim of this systematic review is to gather and analyze studies that assessed the effects of strength training on different variables related to cross-country skiing performance.”

REVIEWER 1

Minor comments:

E.g. Line 38 and beyond: VO2max and VO2peak – I recommend using subscript.

AUTHOR´S

Thank you for your input. We have used subscript for both terms throughout the document.

REVIEWER 1

Line 275: extra text: 5. Conclusions

AUTHOR´S

Excess words have been deleted.

REVIEWER 1
The aim of this systematic review: Effects of strength training on cross-country skiing performance. A systematic review” is to gather and analyze the effect of strength training on different variables related to performance in cross country skiing.
The article is interestingly written and provides useful information for practice. The strength is the gathering of publications dealing with the topic, which is described and implemented with precision. However, there are facts that need to be improved. In particular, the second objective: the analysis of the effect of strength training is not, in my opinion, sufficiently described in the methodology, it should also be given more attention in the results section.
Major comments:
If the aim of the thesis is to analyze the effect of strength training and not to analyze publications dealing with the effect of strength training, it is necessary to supplement the thesis.
The work analyses in detail the quality of already published articles on the topic (Methodological Quality Assessment), but the methodology part does not specify what methodology was used for the analysis of the effect of strength training on different variables related to performance in cross country skiing.
AUTHOR´S

Thank you for your contributions. It is true that there is no calculation of the effect of resistance training on cross-country skiing performance. This is due to the large number of different variables used in the articles included in the review. More than one calculation would be necessary and, above all, given the number of variables, the conclusions would lose reliability, so it was decided to modify the objective and write it as follows: 

“Taking into account the different results found in the literature, the aim of this systematic review is to gather and analyze studies that assessed the effects of strength training on different variables related to cross-country skiing performance.”
REVIEWER 1
Minor comments:
E.g. Line 38 and beyond: VO2max and VO2peak – I recommend using subscript.
AUTHOR´S
Thank you for your input. We have used subscript for both terms throughout the document. 
REVIEWER 1
Line 275: extra text: 5. Conclusions
AUTHOR´S

Excess words have been deleted.

Reviewer 2 Report

This systematic review was well structured to analyze the primary topic  

The selecion of databases (Pubmed, Scopus, Web of Science and Cochrane Library) were sufficient to produce a rather complete review

Interesting and actual the references about performance concerning double-poling.

Future studies should include a wider intervention which analyze a more varied sample.

Important in the review developing, in this type of sport and its derivates, underline the aerobic capacity,  including maximal oxygen uptake (VO2max) and peak oxygen uptake (VO2peak) that are basics the " physiologic and genetic strcture of cross-country skying.

Author Response

REVIEWER 2
This systematic review was well structured to analyze the primary topic  
The selecion of databases (Pubmed, Scopus, Web of Science and Cochrane Library) were sufficient to produce a rather complete review
Interesting and actual the references about performance concerning double-poling.
Future studies should include a wider intervention which analyze a more varied sample.
Important in the review developing, in this type of sport and its derivates, underline the aerobic capacity,  including maximal oxygen uptake (VO2max) and peak oxygen uptake (VO2peak) that are basics the " physiologic and genetic strcture of cross-country skying.
AUTHOR´S

Thank you very much for your comment.

Reviewer 3 Report

Introduction

lines 64-65: This misrepresents the results of Skattebo et al.'s study; what's written here is that there were no improvements at all; many things improved, even though most of the things that improved showed statistically similar improvement between the two groups. Please rewrite for accuracy. Similarly, the prior sentence indicates a similar no-benefit reported in citations 17-19, but all of those seem to have found some improvement

line 56: after the above comment, it seems that "the results obtained to date do not show the same direction" may also misrepresent the data---it seems that there is some benefit to strength training, even if that benefit is not consistently of the same magnitude across studies and in different populations

Methods

lines 87-88: Can you please clarify if the control/comparison group needed to be non-exercise control, or some kind of "usual-care" control (which may be skiing training without added resistance training; you can phrase that some other way than usual-care if that term doesn't make sense to you)

Results

general comment: while obviously you don't want to have a lot of redundancy between the body text of the manuscript and the information presented in the tables, there are many areas where a little more detail is needed in the text to make the point clear; I have tried to highlight them in the below comments, but please double check the whole section

line 132: Unsure how you are using the word carried in this sentence

line 133: Unsure what this means: "replaced the 27% of total endurance training by explosive type strength training"

line 144-145: "the effects of resistance training on performance"--in this sentence: performance of what?

line 146: average power as measured by what?

line 146: maximal power as measured by what?

line 153: Grammatically, when it says "The study by Ofsteng...", it seems like it is referring back to the prior sentence, but Ofsteng is citation #24, and the prior sentence does not include that citation

line 155:  when it says between groups, is that between a strength training group and a control group in all those studies? Also, for this whole paragraph (and other places in the whole manuscript), what are the control groups? I assume it's not people who did no exercise.

lines 156-158: Are the groups that compared different strength training groups, for example a group that lifted at 50%1RM vs 80%1RM? Or is this like on line 152-154 where it was a strength training group versus some kind of control group?

line 159: no changes in performance of what? I know the paragraph is about strength, but you should specify if Skattebo also tested 1RM or something else

Discussion

Lines 204-210: How does this strength-technique relationship apply to the results of the Skattebo et al study?

lines 229-230: I'm unsure of how endurance training is being used here--are you talking about resistance training with acute variables characteristic of endurance phase, or are you talking about aerobic training of skiing?

line 275-276: the section header for conclusion is duplicated here in line with the discussion text

Conclusion

Lines 281-282: I don't think the conclusion, "the ability to transfer maximal strength gains to the locomotion of the different techniques seems
to be decisive
" is supported by the results; earlier you said Skattebo's results alluded to that, but you presented no data that anyone measured technique or did motion analysis

line 282: When you write, ", the rate of strength development", do you mean how fast someone increases their maximal strength, for example, adding 10 kg to their 1RM in 8 weeks instead of 12 (a chronic adaptation), or did you mean rate of force development acutely, for example generating peak force production after the 4th poling stroke instead of the 3rd?

lines 284-285: What limits are suggested by this data?

Lines 288-290: While the sentence " is important to underline that the strength exercises must be movement-specific in order for strength gains to translate into improved performance," makes sense, this hasn't really been presented--the results did not have a section that compared motion specific training programs versus general strength training programs

Table 2: The results arrows don't line up with the outcomes

Table 2: The interventions are inconsistently reported; unsure if that is because the original articles were inconsistent or if not all details are getting transcribed.

Table 2: inconsistent formatting of group membership; sometimes it's IG: 9/CG:10 and sometimes it's 9 IG/7 CG

Author Response

REVIEWER 3
Introduction
lines 64-65: This misrepresents the results of Skattebo et al.'s study; what's written here is that there were no improvements at all; many things improved, even though most of the things that improved showed statistically similar improvement between the two groups. Please rewrite for accuracy. Similarly, the prior sentence indicates a similar no-benefit reported in citations 17-19, but all of those seem to have found some improvement
line 56: after the above comment, it seems that "the results obtained to date do not show the same direction" may also misrepresent the data---it seems that there is some benefit to strength training, even if that benefit is not consistently of the same magnitude across studies and in different populations
AUTHOR´S

Thank you for your contributions. In order to facilitate the reviewer's reading, all modifications to the manuscript have been written in blue font. Lines 62-66 have been rewritten as follows:

On the other hand, various studies did not observe significant improvements in VOâ‚‚max after a period of strength training [17-18]. Furthermore, Carlsson et al. [19] compared strength training versus ski ergometer training and found no significant between-group differences for gross efficiency, upper-body strength and time to exhaustion (TTE).

In addition, the sentence in line 56 has also been changed. What we wanted to say was that there is no consensus about which variables or parameters are improved with strength training. As described in the following lines, some studies showed improvements in VOâ‚‚max, strength and work economy, while others find no significant between-group improvements for these same variables.

“… there seems to be no solid agreement about which performance-related variables are enhanced by strength training.”

REVIEWER 3
Methods
lines 87-88: Can you please clarify if the control/comparison group needed to be non-exercise control, or some kind of "usual-care" control (which may be skiing training without added resistance training; you can phrase that some other way than usual-care if that term doesn't make sense to you)
AUTHOR´S
Thank you for your input. The following clarification has been added:
“… performing traditional cross-country ski training (mostly aerobic training and some muscular endurance).”
REVIEWER 3
Results
general comment: while obviously you don't want to have a lot of redundancy between the body text of the manuscript and the information presented in the tables, there are many areas where a little more detail is needed in the text to make the point clear; I have tried to highlight them in the below comments, but please double check the whole section
AUTHOR´S

Thank you for your contributions.

REVIEWER 3
line 132: Unsure how you are using the word carried in this sentence
AUTHOR´S

Thank you for your observation. The word “carried” has been replaced by “conducted”.

REVIEWER 3
line 133: Unsure what this means: "replaced the 27% of total endurance training by explosive type strength training"
AUTHOR´S
Thank you for your comment. In order to make the article easier to understand, we have modified the sentence.
“All studies conducted maximal strength training [13-17,19,20,24], except Mikkola et al. [18], who implemented explosive resistance training.”
REVIEWER 3
line 144-145: "the effects of resistance training on performance"--in this sentence: performance of what?
AUTHOR´S

Thank you for your comment. In order to make the article easier to understand, we have modified the sentence.

“In addition, several studies analyzed directly the effects of resistance training on cross-country skiing performance,”
REVIEWER 3
line 146: average power as measured by what?
AUTHOR´S

Thank you for your input. We have specified which test was used to calculate the average power.

“…during the 5min double poling test”

REVIEWER 3
line 146: maximal power as measured by what?
AUTHOR´S

Thank you for your comment. We have specified which test were used to calculate the maximal power.

“measured by the 20-second test [20] or by an incremental test on a double poling ergometer [25].”

REVIEWER 3
line 153: Grammatically, when it says "The study by Ofsteng...", it seems like it is referring back to the prior sentence, but Ofsteng is citation #24, and the prior sentence does not include that citation
AUTHOR´S

Thank you for your comment. We have changed the beginning of the sentence as follows:

“Furthermore, Ofsteng…”

REVIEWER 3
line 155:  when it says between groups, is that between a strength training group and a control group in all those studies? Also, for this whole paragraph (and other places in the whole manuscript), what are the control groups? I assume it's not people who did no exercise.
AUTHOR´S

Thank you for your observation. The control groups of all the studies gathered in this review included well-trained cross-country skiers who continued with their previous training program. Traditional cross-country ski training is mostly based on aerobic training and sometimes includes muscular endurance, but not maximal strength. Table 2 specifies the methodologies used in the different studies, and in line 88, the methodology makes specific reference to the control groups.

REVIEWER 3
lines 156-158: Are the groups that compared different strength training groups, for example a group that lifted at 50%1RM vs 80%1RM? Or is this like on line 152-154 where it was a strength training group versus some kind of control group?
AUTHOR´S

Most of the participants were well-trained athletes whose performance would be negatively affected if they did not do any training. Therefore, the difference between the control group and the experimental group is that the control group did not do maximal strength or explosive strength training. This applies to all the research collected in the review.

REVIEWER 3
line 159: no changes in performance of what? I know the paragraph is about strength, but you should specify if Skattebo also tested 1RM or something else
AUTHOR´S

Thank you for your comment. The first part of the paragraph referred to the maximal strength. The second part of the paragraph, instead, mentions the studies that have analyzed the differences in cross-country skiing performance. Most of the studies used different protocols, although all of them were performed on double poling ergometers. Consequently, we have added the following sentences:

“on double poling ergometer”

“measured as the power output on the 20-s test.”

REVIEWER 3
Discussion
Lines 204-210: How does this strength-technique relationship apply to the results of the Skattebo et al study?
AUTHOR´S

Thank you for your comment. We have changed the last part of the paragraph as follows:

“It is important to note that although a certain level of strength is required, greater strength does not necessarily translate into improved performance, as endurance capabilities and technique are both crucial factors in minimizing fatigue accumulation [26,27]. Therefore, Skattebo et al. [20] may not have found an improvement in performance because VOâ‚‚max was significantly reduced or even because they did not focus enough on technique to enable athletes to transfer strength gains to the complex movement of skiing”.

REVIEWER 3
lines 229-230: I'm unsure of how endurance training is being used here--are you talking about resistance training with acute variables characteristic of endurance phase, or are you talking about aerobic training of skiing?
AUTHOR´S

Thank you for your comment. In this case resistance training refers to activities such as roller skiing or running, with an intensity below the anaerobic threshold. We have changed the phrase to make it easier to understand.

“as most of the endurance training was perfomed below the anaerobic threshold”.

REVIEWER 3
line 275-276: the section header for conclusion is duplicated here in line with the discussion text
AUTHOR´S

Thank you for your observation. The error has been corrected.

REVIEWER 3

Conclusion
Lines 281-282: I don't think the conclusion, "the ability to transfer maximal strength gains to the locomotion of the different techniques seems
to be decisive" is supported by the results; earlier you said Skattebo's results alluded to that, but you presented no data that anyone measured technique or did motion analysis
AUTHOR´S

Thank you for your feedback. This review includes studies in which, despite improvements in maximal strength, there have been no significant changes in performance. Therefore, one of the conclusions is that an increase in maximal strength does not always translate into improved performance, as it is important to apply those strength gains to cross-country skiing locomotion. However, the sentence has been modified as follows:

“…although gains in maximal strength do not always induce improvements in economy and performance.”

REVIEWER 3
line 282: When you write, "the rate of strength development", do you mean how fast someone increases their maximal strength, for example, adding 10 kg to their 1RM in 8 weeks instead of 12 (a chronic adaptation), or did you mean rate of force development acutely, for example generating peak force production after the 4th poling stroke instead of the 3rd?
AUTHOR´S

Thank you for your comment. By rate of strength development we mean rate of force development. The word “strength” has been changed to “force”.

REVIEWER 3
lines 284-285: What limits are suggested by this data?
AUTHOR´S

Thank you for your question. The limits will depend on each subject, the point is that they produce adaptations without leading to overtraining.

“..are adjusted to the subject so that adaptations occur without excessive fatigue.”

REVIEWER 3
Lines 288-290: While the sentence "is important to underline that the strength exercises must be movement-specific in order for strength gains to translate into improved performance," makes sense, this hasn't really been presented--the results did not have a section that compared motion specific training programs versus general strength training programs
AUTHOR´S

Thank you for your observation. This idea has been mentioned because studies that reported improvements in work economy or even performance have indicated that it is very important for strength exercises to be movement specific. 

REVIEWER 3
Table 2: The results arrows don't line up with the outcomes
AUTHOR´S
Thank you for your observation. The error has been fixed.

REVIEWER 3
Table 2: The interventions are inconsistently reported; unsure if that is because the original articles were inconsistent or if not all details are getting transcribed.
AUTHOR´S

Thank you for your comment. The studies have been described as accurately as in the articles.

REVIEWER 3
Table 2: inconsistent formatting of group membership; sometimes it's IG: 9/CG:10 and sometimes it's 9 IG/7 CG
AUTHOR´S

Thank you for your observation. We have applied the same format for all the studies.

Round 2

Reviewer 1 Report

The authors have edited the text in accordance with the comments, I have no next comment. Congratulations to the authors for publishing the study.

Reviewer 3 Report

Thank you for your diligent response to all reviewer comments. I have no further comments at this stage.